# Is There Bias in the Assessment of Contraindications for Resection? Disparities in the Surgical Management of Early-Stage Esophageal Cancer

**DOI:** 10.3390/diseases13020037

**Published:** 2025-01-30

**Authors:** Christina S. Boutros, Lauren M. Drapalik, Christine E. Alvarado, Aria Bassiri, Jillian Sinopoli, Leonidas Tapias Vargas, Philip A. Linden, Christopher W. Towe

**Affiliations:** Department of Surgery, Division of Thoracic Surgery, University Hospitals Cleveland Medical Center, Cleveland, OH 44106, USA; christina.lauren.drapalik2@uhhospitals.org (L.M.D.); christine.alvarado@uhhospitals.org (C.E.A.); aria.bassiri@uhhospitals.org (A.B.); jillian.sinopoli@uhhospitals.org (J.S.); leonidas.tapias@uhhospitals.org (L.T.V.); philip.linden@uhhospitals.org (P.A.L.); chrisopher.towe@uhhospitals.org (C.W.T.)

**Keywords:** esophageal cancer, disparities, minorities

## Abstract

Background: Resection is considered the standard of care for patients with localized esophageal cancer who are “physiologically fit”. Patients who do not meet this standard are considered contraindicated to receive surgery. We hypothesized that among patients with non-metastatic esophageal cancer, the consideration of contraindication status would vary based on clinical and demographic factors and would vary between institutions. Methods: We identified patients with non-metastatic gastric and esophageal cancer in the National Cancer Database (NCDB) from 2004 to 2018. Patients were categorized into three groups based on surgical treatment: surgical resection (including endoscopic mucosal resection), resection contraindicated, and refusal of resection based on the coding of the “reason for no surgery” data element. Demographic, clinical, and institutional characteristics were compared between the groups using bivariate and multivariate techniques to identify factors associated with contraindicated status. A subgroup analysis of cT1N0M0 patients was also used to assess every institution in the NCDB’s observed–expected ratio for contraindication status. Results: In total, 144,591 patients with non-metastatic disease met inclusion criteria: 124,972 (86%) underwent resection, 13,793 (10%) were contraindicated for resection, and 5826 (4%) refused resection. Contraindication was associated with age, non-Hispanic Black race, socioeconomic status, Charlson–Deyo score, insurance type, institution characteristics, clinical T-stage, and clinical N-stage. There were 9459 patients who were cT1N0M0 and had no co-morbidities. In this cohort, there were more than 1000-fold differences between individual programs regarding observed–expected ratio of contraindication status when adjusting for clinical and demographic characteristics. Conclusions: Variation in the assessment of contraindication status varies dramatically between institutions. Underserved minorities, including age, race, and insurance type, are risk factors for being considered contraindicated. These findings highlight the disparities that exist regarding surgical care of non-metastatic esophageal cancer in the United States.

## 1. Introduction

Esophageal cancer is the sixth leading cause of cancer-related deaths globally and cases are expected to rise. Resection can be curative and is considered the standard of care for patients who are “physiologically fit” with early-stage and locally advanced disease. The NCCN defines locally advanced disease as stage IIb to IIIc and includes tumors that invade regional lymph nodes (N1–3) or local structures (T4 disease) and recommends surgical resection in all patients that are considered “medically fit” [1]. Although, globally, outcomes for surgical resection of esophageal cancer have improved, there continues to be considerable risk of morbidity and mortality following resection. As such, many patients are deemed to be too high risk for surgery and receive inferior treatments.

Patients that are traditionally thought of as being contraindicated for surgical resection are patients with metastatic disease, considered to be very frail, or patients who suffer from comorbid conditions, which would make the risk of surgical resection unacceptable. Some providers may feel that a patient is disqualified based on age alone. Although there is no established age cut-off at which patients are recommended against surgical resection, Hung-Chang Liu et al. describe age 70 as “elderly” for esophageal cancer surgery [2]. Current literature on the determination of “physiologically fit” versus “contraindicated for surgical resection” is sparse. The Society of Thoracic Surgeons guidelines for diagnosis and staging of esophageal cancers recommend a “physiologic workup” which may include pulmonary function tests and cardiovascular risk stratification. However, these guidelines do not include recommendations for the determination of contraindications for surgery [3]. Similarly, the practice guidelines from the Japanese Esophageal Society recommend definitive chemoradiation for “those unable to tolerate surgery” but do not provide criteria to establish this group [4]. Most patients without significant comorbid disease burden, who are physiologically fit and do not have metastatic disease, should then be considered for surgical resection. In the absence of definitive guidelines, the assessment of who should receive surgery is subjective in nature and is open to variations in practice. This subjective assessment is subject to variation in practice, which in turn is an opportunity for bias.

Well-documented disparities exist in esophageal cancer care in terms of outcomes and survival; however, there are few documented studies that have examined disparities in surgical decision-making [5,6,7]. To our knowledge, there are no documented studies that assess the rate at which patients are considered to be contraindicated for surgical resection across institutions. This study seeks to assess the patient and program factors linked to the determination of a patient being deemed unsuitable for resection. We hypothesize that among individuals with non-metastatic esophageal cancer, the likelihood of being classified as contraindicated for surgery will differ based on clinical and demographic characteristics. Additionally, the frequency of contraindication designations is expected to vary between institutions.

## 2. Methods

### 2.1. Data Source

The 2018 version of the gastric and esophageal National Cancer Database (NCDB) was used for this study. The NCDB collects data from the Commission on Cancer-accredited institutions in the United States. This dataset contains information on approximately 70% of new cancer diagnoses each year. The NCDB is maintained by the American College of Surgeons and the American Cancer Society. These societies do not monitor the accuracy of the data reported, the statistical analyses performed, or the conclusions drawn by the authors. The NCDB Participant User File data dictionary contains definitions of variables used in this study.

### 2.2. Patient Cohort

We identified patients with non-metastatic gastric and esophageal cancer from 2004 to 2018. Patients with primary-site gastric and esophageal cancer were included. Patients were excluded if they had clinically confirmed distant metastatic disease (stage IV). Patients were also excluded if their “reason for no surgery” data variable was missing, if they died prior to receiving treatment, or if surgery was not part of the initial treatment plan.

### 2.3. Exposure

Patients were grouped by their surgical treatment based on the data variable “reason for no surgery”: surgical resection (including endoscopic mucosal resection), resection contraindicated, and refusal of resection.

### 2.4. Outcomes

The primary outcome for all patients was their contraindication status. Survival data was also analyzed as a secondary outcome based on surgical intervention, or lack thereof. The NCDB does not include survival data for patients diagnosed in 2018; thus, survival analyses are representative of patients diagnosed in 2004–2017.

### 2.5. Statistical Analysis

Demographic and clinical data were compared between the three groups to identify risk factors for contraindications to surgical resection. Pearson’s chi-squared test was used to compare categorical variables, and the rank sum test was used for continuous variables. To reduce confounding, we performed a multivariable logistic regression to determine the association between clinical and demographic data and being considered a contraindication for surgical resection. Cofactors for the regression were chosen based on backward selection from univariate modeling. Median overall survival was estimated using the Kaplan–Meier method and was compared using the log-rank test. A subgroup analysis of cT1N0M0 patients with a Charlson–Deyo score of 0 was used to create a multivariable logistic model of factors associated with contraindication status. This model was used to assess every institution in the NCDB’s observed–expected ratio for contraindication status. Data are presented as odds ratios and 95% confidence intervals. A *p*-value < 0.05 was used to indicate statistical significance. StataSE v16.1 (Statacorp LLC, College Station, TX, USA) was used for statistical analyses.

### 2.6. Institutional Assurances

This project was exempt from Institutional Review Board approval at our institution due to the deidentified nature of the dataset.

## 3. Results

*Demographic and Clinical Data:* In total, 144,591 patients met the inclusion criteria. 124,972 (86%) underwent surgical resection, 13,793 (10%) were considered contraindicated, and 5826 (4%) refused. The relationship of age to contraindication status is shown in Figure 1. A contraindication was associated with reduced overall survival compared to receipt of surgery and refusal of surgery (*p* < 0.001 by the log-rank test, Figure 2). Several demographic factors were associated with contraindicated status, including age, sex, race, Charlson–Deyo comorbidity score, and facility type (Table 1). Contraindication was also associated with disease characteristics, including esophageal vs. gastric cancer, clinical T-stage, and clinical N-stage.

*Multivariate analysis:* Contraindication status was associated with age (OR 1.07, CI 1.07–1.08), non-Hispanic Black race (OR 1.37, CI 1.23–1.52), low socioeconomic status (OR 1.3, CI 1.13–1.50), and Charlson–Deyo comorbidities (Table 2). Patients with Medicaid were more than twice as likely to be considered contraindicated for surgical resection when compared to their counterparts with private insurance (OR 2.3 CI, 1.99–2.70). Patients who received their care at low-volume institutions were also nearly twice as likely to be considered contraindicated for surgery than those who received care at a high-volume institution (OR 1.8, CI 1.49–2.18). Higher clinical T-stage and clinical N-stage were associated with greater odds of being contraindicated for surgical resection. Comprehensive community programs (OR 1.65, CI 1.24–2.40) were also associated with higher rates of contraindication.

*Subgroup analysis:* There were 9459 patients who were cT1N0M0 and had no Charlson–Deyo co-morbidities. In this cohort, age > 70 (OR 2.24, CI 1.39–3.59), age > 80 (OR 7.31, CI 4.66–11.48), non-Hispanic Black race (OR 1.46, CI 1.11–1.93), Medicaid insurance type (OR 3.22, CI 2.19–4.73), lowest socioeconomic status (OR 1.50, CI 1.05–2.13), comprehensive community programs (OR 1.73, CI 1.08–2.77), and treatment at low-volume centers (OR 1.70, CI 1.20–2.40) were all independently associated with contraindication status (Table 3). Based on this model, we created a model of institutional variation in contraindication status (Figure 3). There were dramatic differences between individual institutions with >1000-fold variation in the determination of contraindication status.

## 4. Discussion

Due to the lack of true standardization of operative contraindication criteria, the decision for surgery is influenced by provider perception. Where there is no standardization of care, there is room for variation in care and bias, which can impact marginalized communities [4,5,8,9,10]. Our study underscores the wide variation across individual programs across the United States and demonstrates that underserved minorities are more likely to be considered contraindicated for surgery, even among a cohort of patients without Charlson–Deyo comorbidities.

This study showed that contraindicated status was more likely among patients with age > 70, non-Hispanic Black race, Medicaid insurance type, lowest socioeconomic status, and treatment at low volume centers. Other studies have noted an underutilization of esophagectomy in patients who are Black or with lower socioeconomic status [11,12]. Merrit et al. utilized the NCDB to study the differences in provider recommendations for esophagectomy between Black and White patients who have esophageal cancer that is operable (stage I, II, and III). They found that non-Hispanic Black patients have three times greater odds of not being recommended esophagectomy [13]. Their study, like ours, adjusted for patient age, income, insurance status, gender, TNM clinical stage, urban/rural location, Charlson–Deyo score, and healthcare facility type. As a result of a disparity in recommendations by health care providers, Black patients had a significantly worse median and overall survival compared to White patients. They noted that even when healthcare providers recommended esophagectomy, Black patients were less likely to undergo the procedure.

Age is another patient factor that influences the decision to perform esophagectomy for esophageal cancer. Esophagectomy in the elderly remains a controversial area of continued discussion with mixed data on outcomes. Some studies show that age is associated with worse in-hospital mortality and morbidity [14,15,16,17,18], whereas other studies show similar outcomes when compared to a younger cohort [19]. This analysis suggests the rate of surgical contraindication increases dramatically at age 70. Although age has classically been used as a surrogate measure of frailty, newer data suggests that age alone is a poor predictor of ill-health and that other measures are better predictors of who would fare well after esophagectomy. Tang et al. developed the Esophageal Vitality Index, which measures grip strength, 30 s chair sit-stands, 6 min walk, and psoas muscle area to height ratio to predict post-esophagectomy mortality and morbidity [20]. Using age as a surrogate for frailty can unnecessarily exclude patients from receiving a potentially curative intervention.

This study also demonstrated dramatic differences in contraindication status between institutions. The variation in practice at the determination of contraindication may be able to be addressed through the centralization of esophageal cancer care at high-volume institutions. Schlottmann et al. [21] support this sentiment and find that centralization of esophageal cancer surgery has resulted in improving mortality and outcomes for all patients. They analyzed the Nationwide Inpatient Sample from 2000 to 2014 and found that non-white patients and patients of low household income also demonstrated a reduction in hospital mortality after esophagectomy during the study period. It is known that for complex operations like pancreatectomy, pneumonectomy, and esophagectomy, there is a relationship between facility volume and a reduction in patient mortality, where patients who have their care at higher volume institutions have better outcomes in terms of both morbidity and mortality. Access to high-volume institutions, however, is skewed. Patients who live in rural areas are more likely to undergo esophagectomy in a program with low surgical volume [1] than those who live in an urban setting. African Americans are less likely to undergo esophagectomy for esophageal cancer compared with White patients, and the esophageal cancer mortality rate for Black patients is twice that of White patients [1]. Our data showed that there is a greater than 1000-fold variation in contraindication status across institutions in the United States and that low facility volume was independently associated with greater odds of contraindication, and as such, centralization of esophageal cancer care at high volume intuitions may be crucial in achieving equal outcomes of care and recommendations for patients.

### Limitations

There are several limitations to acknowledge when interpreting our findings. One of the most significant is that the NCDB does not include detailed information on patient comorbidities, performance status, re-operative status, disease burden, access to care, or frailty metrics, which may have introduced bias into our results. Furthermore, this study was a retrospective analysis based on deidentified data from a national database. We are unable to address potential inaccuracies or missing data within the database. The NCDB only collects information on approximately 70% of newly diagnosed cancers and primarily includes data from hospitals accredited by the Commission on Cancer, potentially excluding patients diagnosed or treated at non-accredited facilities. Additionally, the large sample size in this study may lead to the identification of statistically significant differences that may have limited clinical significance. Despite these limitations, we believe that there is notable variation in practice across institutions in the United States and how marginalized communities are negatively affected.

## 5. Conclusions

Approximately 10% of patients with esophageal cancer are seen as contraindicated for resection. Variation in the assessment of contraindication status varies more than 1000 times between institutions. Underserved minorities, including non-Hispanic Black race, age over 70, and Medicaid insurance are risk factors for being considered contraindicated. These findings highlight the significant disparities that exist regarding surgical care of non-metastatic esophageal cancer in the United States and underscore existing arguments for regionalization of care.

## Figures and Tables

**Figure 1 diseases-13-00037-f001:**
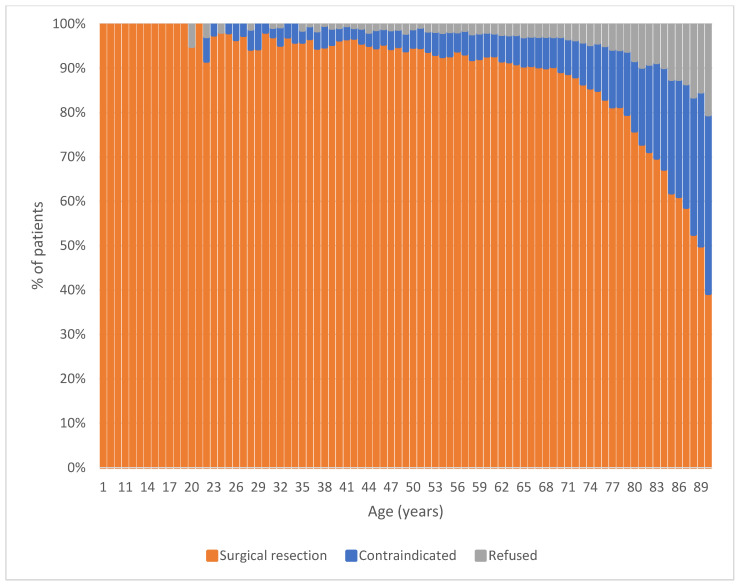
The relationship between age and receipt of surgery, refusal, and being considered contraindicated for surgery among patients with non-metastatic gastric and esophageal cancer in the National Cancer Database (2004–2018).

**Figure 2 diseases-13-00037-f002:**
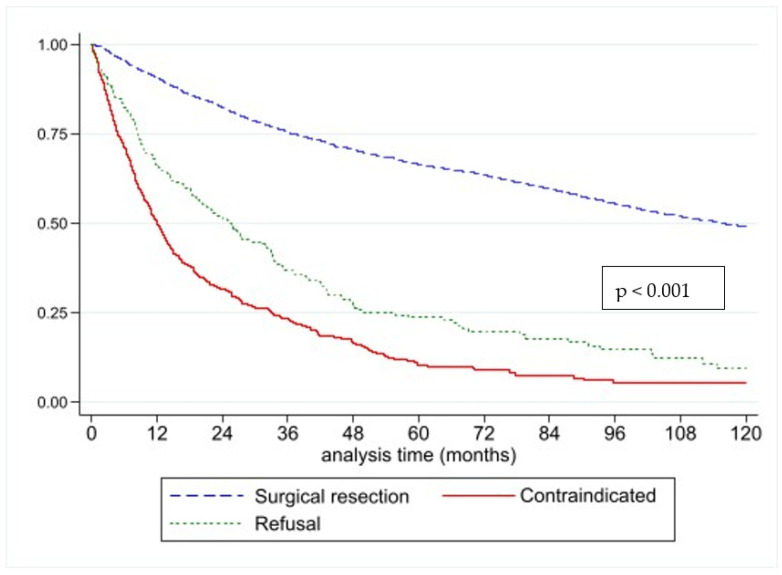
Kaplan–Meier survival estimate for patients with gastric and esophageal cancer comparing receipt of surgery, refusal of surgery, and contraindicated status.

**Figure 3 diseases-13-00037-f003:**
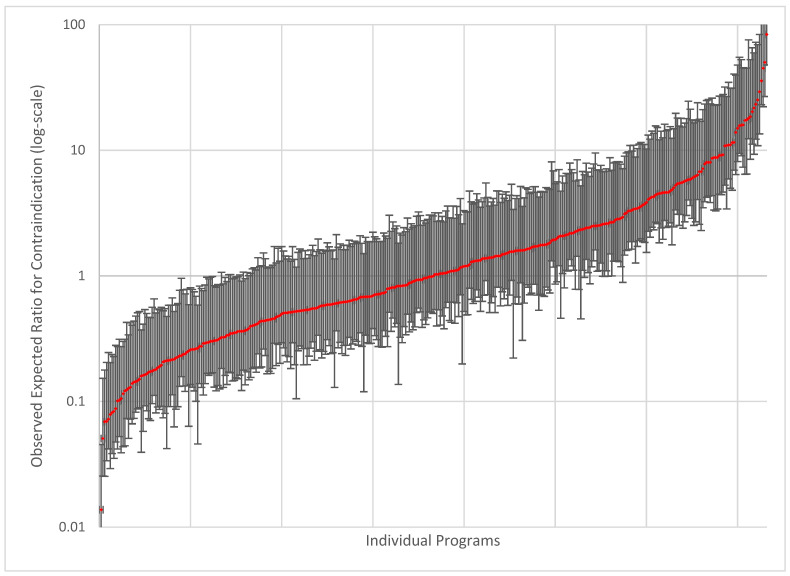
Caterpillar plot of odds of a cT1N0M0 patient without Charlson–Deyo comorbidity being considered contraindicated for surgery at a given institution in the National Cancer Database (2004–2018). The red line is the actual observed:expected ratio and the black lines are 95% Confidence interval.

**Table 1 diseases-13-00037-t001:** Clinical and demographic factors associated with surgical resection, contraindication status, and refusal of surgery for patients with non-metastatic esophageal cancer and gastric cancer in the National Cancer Database. Data are presented as n (%) and compared using chi-square (The colors represent different categories).

	Surgical	Contraindicated	Refused	*p*-Value
**Sex**				0.001
**Female**	38,272 (31%)	4474 (32%)	2140 (37%)	
**Male**	86,700 (69%)	9319 (68%)	3686 (63%)	
**Race**				0.001
**NH White**	87,603 (70%)	10,241 (74%)	4066 (70%)	
**NH Black**	13,823 (11%)	1652 (12%)	859 (15%)	
**Hispanic**	9338 (7%)	657 (5%)	265 (4%)	
**Other/Unknown**	14,208 (12%)	1243 (9%)	636 (11%)	
**Charlson–Deyo Comorbidity Index**				0.001
**Charlson–Deyo 0**	85,647 (69%)	7925 (57%)	3727 (64%)	
**Charlson–Deyo 1**	27,979 (22%)	13,793 (24%)	1286 (22%)	
**Charlson–Deyo 2**	7779 (6%)	1464 (11%)	499 (9%)	
**Charlson–Deyo >3**	3567 (3%)	1159 (8%)	314 (6%)	
**Facility Type**				0.001
**Community Cancer Program**	5788 (5%)	1168 (9%)	530 (9%)	
**Comprehensive Community Program**	38,266 (31%)	5423 (40%)	2417 (42%)	
**Academic/Research Program**	54,490 (45%)	4196 (31%)	1736 (30%)	
**Integrated Network Cancer Program**	23,331 (19%)	2897 (21%)	5805 (19%)	
**Insurance Type**				0.001
**Not Insured**	3116 (2%)	337 (2%)	131 (2%)	
**Private Insurance**	48,587 (39%)	2315 (17%)	908 (16%)	
**Medicaid**	8051 (6%)	836 (6%)	319 (6%)	
**Medicare**	61,186 (49%)	9851 (71%)	4292 (74%)	
**Other Government**	1738 (1%)	255 (2%)	82 (1%)	
**Unknown Insurance Status**	2294 (2%)	199 (2%)	94 (2%)	
**Socioeconomic Status**				0.001
**SES 0**	11,323 (10%)	1302 (10%)	625 (12%)	
**SES 1**	12,327 (11%)	1429 (11%)	617 (11%)	
**SES 2**	16,008 (14%)	1869 (14%)	749 (14%)	
**SES 3**	18,195 (16%)	2047 (16%)	924 (17%)	
**SES 4**	18,750 (16%)	2226 (17%)	862 (16%)	
**SES 5**	17,576 (15%)	1836 (14%)	766 (14%)	
**SES 6**	20,671 (18%)	2209 (17%)	5424 (16%)	
**Facility Surgical Volume**				0.001
**Low**	32,753 (26%)	5376 (39%)	2378 (40%)	
**Medium**	42,665 (34%)	4699 (34%)	2078 (36%)	
**High**	49,524 (40%)	3718 (27%)	1370 (24%)	
**Type of Cancer**				0.001
**Gastric Cancer**	83,731 (67%)	6341 (45%)	3075 (53%)	
**Esophageal Cancer**	41,241 (33%)	7452 (54%)	2751 (47%)	
**Clinical T-Stage**				0.001
**0**	1563 (2%)	140 (2%)	85 (2%)	
**1**	17,281 (21%)	1862 (21%)	904 (25%)	
**2**	20,395 (25%)	1981 (22%)	841 (23%)	
**3**	39,261 (48%)	4208 (47%)	1686 (46%)	
**4**	2982 (4%)	722 (8%)	133 (4%)	
**Clinical N-Stage**				0.001
**0**	62,543 (64%)	6690 (59%)	3047 (66%)	
**1**	27,731 (28%)	3610 (32%)	1268 (27%)	
**2**	6819(7%)	854 (8%)	283 (6%)	
**3**	1348 (1%)	214 (2%)	38 (1%)	
**Facility Location**				
**New England**				

**Table 2 diseases-13-00037-t002:** Multivariable logistic regression of factors associated with being considered contraindicated for surgery among patients with non-metastatic gastric and esophageal cancer in the National Cancer Database (2004–2018) (The colors represent different categories).

Cofactor	Odds Ratio	Confidence Interval	*p*-Value
**Age**			
**40-**	0.65	0.23–1.85	0.42
**45-**	0.49	0.22–1.01	0.072
**50-**	0.79	0.47–1.37	0.41
**55-**	1.2	0.79–2.0	0.34
**60-**	*ref.*		
**65-**	1.14	0.72–1.79	0.58
**70-**	2.24	1.40–3.60	0.001
**75-**	3.26	2.11–5.04	0.001
**80-**	7.32	4.66–11.49	0.001
**85-**	15.19	9.70–23.79	0.001
**90-**	25.74	14.76–44.87	0.001
**Sex**			
**Male**	*ref.*		
**Female**	0.95	0.79–1.13	0.55
**Race**			
**Non-Hispanic White**	*ref.*		
**Non-Hispanic Black**	1.47	1.11–1.94	0.006
**Hispanic**	0.57	0.37–0.89	0.014
**Other/Unknown**	0.79	0.59–1.07	0.13
**Facility Type**			
**Community Cancer Program**	1.48	0.93–2.36	0.096
**Comprehensive Community**	1.65	1.24–2.40	0.001
**Academic**	*ref.*		
**Integrated Network Cancer Network**	1.22	0.86–1.71	0.26
**Insurance Status**			
**Not Insured**	1.49	0.66–3.35	0.33
**Private Insurance**	*ref.*		
**Medicaid**	3.2	2.19–4.73	0.001
**Medicare**	1.33	0.96–1.84	0.083
**Other Government**	1.48	0.70–3.13	0.31
**Unknown Insurance Status**	0.97	0.37–1.76	0.94
**Socioeconomic Status**			
**0**	1.4	1.05–2.213	0.027
**1**	1.2	0.90–1.75	0.18
**2**	1.1	0.80–1.54	0.54
**3**	1.18	0.89–1.61	0.29
**4**	1.03	0.77–1.38	0.83
**5**	1.32	0.98–1.76	0.06
**6**	*ref.*		
**Clinical T-Stage**			
**0**	*0.75*	0.69–0.92	0.004
**1**	*ref.*		
**2**	*0.88*	0.81–0.97	0.009
**3**	*0.98*	0.90–1.07	0.697
**4**	*2.5*	2.19–2.87	0.001
**Clinical N-Stage**			
**0**	*ref.*		
**1**	*1.33*	1.25–1.42	0.001
**2**	*1.38*	1.25–1.53	0.001
**3**	*1.78*	1.43–2.21	0.001
**Type of Cancer**			
**Gastric Cancer**	*ref.*		
**Esophageal Cancer**	3.6	2.95–4.40	0.001
**Facility Volume**			
**Low**	1.7	1.20–2.40	0.003
**Medium**	1.36	1.05–1.85	0.046
**High**	*ref.*		
**Charlson–Deyo Comorbidity Index**			
**0**	*ref.*		
**1**	*1.12*	1.04–1.20	0.002
**2**	*2.07*	1.91–2.24	0.001
**Facility Location**			
**New England**	*1.36*	1.10–1.69	0.005
**Middle Atlantic**	*ref.*		
**South Atlantic**	*0.82*	0.67–1.00	0.051
**East North Central**	*1.05*	0.81–1.35	0.734
**East South Central**	*0.73*	0.56–0.96	0.022
**West North Central**	*0.98*	0.75–1.30	0.921
**West South Central**	*0.80*	0.58–1.11	0.182
**Mountain**	*1.02*	0.74–1.40	0.901
**Pacific**	*0.73*	0.59–0.91	0.006

**Table 3 diseases-13-00037-t003:** Subgroup multivariable regression: regression of factors associated with being considered contraindicated for surgery among patients with cT1N1M0 gastric and esophageal cancer and no Charlson–Deyo comorbidities in the National Cancer Database (2004–2018) (The colors represent different categories).

Factors Associated with Contraindication	Odds Ratio	Confidence Interval	*p*-Value
**Age**			
**45-**	0.46	0.14–1.48	0.19
**50-**	0.91	0.41–2.01	0.82
**55-**	1.1	0.54–2.28	0.77
**60-**	*ref.*		
**65-**	1.28	0.66–2.46	0.46
**70-**	2.33	1.16–4.60	0.017
**75-**	3.48	1.84–6.61	0.001
**80-**	8.09	3.99–16.41	0.001
**85-**	28.70	12.95–63.58	0.001
**90-**	47.82	17.24–132.87	0.001
**Sex**			
**Male**	*ref.*		
**Female**	1.11	0.77–1.61	0.56
**Type of Cancer**			
**Gastric Cancer**	*ref.*		
**Esophageal Cancer**	3.60	2.95–4.40	0.001
**Race**			
**Non-Hispanic White**	*ref.*		
**Non-Hispanic Black**	1.92	1.05–3.52	0.034
**Hispanic**	0.22	0.05–0.95	0.043
**Other/Unknown**	1.12	0.62–2.02	0.700
**Facility Type**			
**Community Cancer Program**	1.56	0.67–3.66	0.299
**Comprehensive Community**	1.73	1.08–2.77	0.022
**Academic**	*ref.*		
**Integrated Network Cancer Network**	1.03	0.62–1.75	0.88
**Insurance Status**			
**Not Insured**	2.68	0.94–7.67	0.066
**Private Insurance**	*ref.*		
**Medicaid**	2.29	1.07–4.90	0.031
**Medicare**	1.43	0.86–2.65	0.158
**Other Government**	2.33	0.73–7.44	0.152
**Unknown Insurance Status**	1.41	0.34–5.77	0.63
**Socioeconomic Status**			
**0**	1.97	0.94–4.11	0.072
**1**	1.92	1.09–3.36	0.023
**2**	1.52	0.86–2.65	0.143
**3**	1.19	0.68–2.07	0.528
**4**	0.67	0.40–1.15	0.150
**5**	1.27	0.74–2.16	0.376
**6**	*ref.*		
**Facility Volume**			
**Low**	2.44	1.43–4.17	0.001
**Medium**	1.42	0.85–2.33	0.173
**High**	*ref.*		
**Facility Location**			
**New England**	*1.17*	0.80–1.71	0.417
**Middle Atlantic**	*ref.*		
**South Atlantic**	*0.90*	0.65–1.24	0.517
**East North Central**	*0.92*	0.59–1.42	0.710
**East South Central**	*0.43*	0.25–0.76	0.004
**West North Central**	*0.79*	0.51–1.22	0.291
**West South Central**	*0.90*	0.48–1.67	0.738
**Mountain**	*0.99*	0.56–1.75	0.978
**Pacific**	*0.61*	0.42–0.89	0.012

## Data Availability

Data are contained within the article and can be obtained from the authors upon reasonable request.

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
