# Peer review of "Is There Bias in the Assessment of Contraindications for Resection? Disparities in the Surgical Management of Early-Stage Esophageal Cancer"

_diseases, 2025, doi:10.3390/diseases13020037_

Round 1
Reviewer 1 Report
Comments and Suggestions for Authors
This is an important and well written manuscript and one has to congratulate the authors for their work.
Unfortunately, the topic has been published multiple times with the same message. Low income, low socioeconomic background, race equals less access to the right institutions. Boutros et al. fail to provide any new insights.
Moreover the data quality of the National Cancer Database is limited as stated by the authors themselves.
I think the paper can be well described as stated by the author themselves: "Our study underscores the wide variation across individual programs across the United States and demonstrates that underserved minorities are more likely to be considered contraindicated for surgery "
Author Response
We thank the reviewer for acknowledging the importance and clarity of our manuscript. While it is true that disparities in access to surgical care have been previously described, our study specifically addresses the subjective assessment of surgical contraindications in early-stage esophageal cancer, which remains underexplored. By highlighting how institutional practices vary and how underserved minorities are disproportionately affected by these variations, our findings provide granular insights into a specific aspect of healthcare disparities not previously documented in this detail.
We appreciate the reviewer’s comment regarding the limitations of the National Cancer Database (NCDB). While we acknowledge that the NCDB has inherent limitations, it remains one of the most comprehensive sources for analyzing patterns of care across the United States, particularly for minority and underserved populations. To address this concern, we have discussed this in the limitations section. We believe this transparency reinforces the robustness of our conclusions despite these constraints.
We appreciate the reviewer highlighting this key conclusion of our work. This statement encapsulates our primary findings and underscores the critical need for targeted interventions to address institutional variations and reduce disparities.
Reviewer 2 Report
Comments and Suggestions for Authors
This is a retrospective study using the National Cancer Database which collects data from many Commission on Cancer-accredited institutions in the US. The study demonstrated the disparities that exist and differences among institutions. The abstract is clearly written with the purpose and methods described well. The results are noted, but more information about the contraindications would be helpful. The differences among the institutions are clear, but the results indicating the disparities are not as clear. Can the disparity factors be made clearer in the results to support the conclusions that are made? There is adequate background related to the issue of who is appropriate for surgical consideration. There is good discussion of the current guidelines internationally. The aim of the study was clearly stated as well as the hypotheses. The methods are clear including the data source, inclusion / exclusion criteria, grouping of patients, and data used in the study. The statistical analyses are described well. Since there are many analyses included in this study, the authors might consider a Bonferroni correction. Results are clearly presented in a number of figures and tables. The discussion is complete with discussion of the results and comparisons to similar studies. The discussion of the issue of age is important and well stated. The variation among institutions is well developed. The limitations are clearly stated. It could be noted that due to the large number of patients in the samples, small differences resulting in many significant differences were found. The conclusions follow well from the results.
Author Response
We appreciate the reviewer’s suggestion to provide more detailed information about the contraindications for surgery. Unfortunately, due to the limitations of the NCDB, the reason_for_no_surgery variable only indicates whether surgery was deemed contraindicated but does not provide specific reasons for the contraindication. To address this limitation, we narrowed our cohort to patients who, based on the available variables in the NCDB (e.g., stage, Charlson-Deyo comorbidity score), would likely meet criteria for resection under normal circumstances. This approach allowed us to focus on patients who were presumed to be surgical candidates. From this refined cohort, we examined demographic and socioeconomic factors, as well as institutional variation, to assess how these variables influenced the determination of contraindication status.
Thank you for this valuable feedback. In the Results section (lines 143 – 148 ) we stated that “There were 9,459 patients who were cT1N0M0 and had no Charlson-Deyo co-morbidities. In this cohort, age > 70 (OR 2.24, CI 1.39 – 3.59), age > 80 (OR 7.31, CI 4.66-11.48), non-Hispanic Black race (OR 1.46, CI 1.11-1.93), Medicaid insurance type (OR 3.22, CI 2.19-4.73), lowest socioeconomic status (OR 1.50, CI 1.05-2.13), Comprehensive Community programs (OR 1.73, CI 1.08 – 2.77) and treatment at low volume centers (OR 1.70 CI 1.20-2.40) were all independently associated with contraindication status.(Table 2).” Which details a breakdown of the disparity factors, including income, insurance status, and race/ethnicity, and how these variables were associated with surgical contraindications. This clarification directly supports the conclusions and emphasizes the multifaceted nature of the observed disparities. Due to the de-identified nature of the NCDB we cannot determine individual hospital characteristics.
We appreciate the reviewer’s suggestion regarding the potential use of a Bonferroni correction. However, we respectfully believe that applying this correction is not necessary in our study because we are testing a single overarching hypothesis: that underserved minorities are more likely to be deemed contraindicated for surgery. The multiple statistical tests conducted in this study serve to evaluate different facets of this singular hypothesis, rather than testing multiple independent hypotheses. This approach is standard in scientific writing, particularly in studies exploring complex phenomena like healthcare disparities, and does not constitute the type of multiple comparisons that Bonferroni corrections are designed to address.
Additionally, we have not encountered this level of granularity in prior studies of surgical contraindications, making our methodology uniquely comprehensive. Applying a Bonferroni correction in this context could potentially mask important findings by being overly conservative. We hope this clarification is satisfactory, but we are happy to provide further details if needed.
We thank the reviewer for highlighting this important point. We have added a note in the Limitations section (lines 226-227) to acknowledge that the large sample size may amplify the detection of statistically significant differences, some of which may have limited clinical relevance. This addition enhances the transparency and context of our findings.
We are grateful for the reviewer’s positive feedback on these aspects of our manuscript.
Reviewer 3 Report
Comments and Suggestions for Authors
This is a very well written article. It is about the surgical management of the early-stage esophageal cancer (T1N0M0), which is a very interesting and discussed topic, with a huge a number of cases in many countries.
This study included 144591 patients, treated between 2004 and 2018. 86 % of them underwent resection, 10 % were contraindicated and 4 % refused resection.
The authors compared demographic, clinical, and institutional characteristics of these patients from the National Cancer Database (NCDB). The study reveals that the assessment of contraindication varies dramatically (more than 1,000 times) between institutions and that there are significant disparities regarding surgical care of non-metastatic esophageal cancer in the United States.
I recommend the publication of this article in Diseases.
Author Response
We sincerely thank the reviewer for their thoughtful and supportive comments. We are pleased that the reviewer found the manuscript to be well-written and that the findings were impactful and relevant to the ongoing discussion about surgical management of early-stage esophageal cancer. We greatly appreciate the recommendation for publication and value the positive feedback on the clarity and significance of our study.
Round 2
Reviewer 1 Report
Comments and Suggestions for Authors
Thank you for your response
Author Response
Comment from reviewer: Thank you for your response
Response: You're welcome. Thank you for taking the time to review our work.